# Enhancement of Photocatalytic Activities with Nanosized Polystyrene Spheres Patterned Titanium Dioxide Films for Water Purification

**Hyeon Jin Seo** [1]**, Ji Won Lee** [2]**, Young Hoon Na** [1] **and Jin-Hyo Boo** [1,2,*]

[1]    Department of Chemistry, Sungkyunkwan University, Suwon 16419, Korea; qpzwoxeic@skku.edu (H.J.S.); yhna@hanmail.net (Y.H.N.)

[2]    Institute of Basic Science, Sungkyunkwan University, Suwon 16419, Korea; ljw9917@naver.com

*    Correspondence: jhboo@skku.edu; Tel.: +82-31-290-7072

**Abstract:** For environmental applications, such as water and air purification utilizing photocatalysts, we synthesized patterned titanium dioxide ($TiO_2$) thin films using polystyrene (PS) spheres. This was primarily done to enhance the surface area and photocatalytic activities. $TiO_2$ thin films were deposited on silicon wafers attached to variously sized PS spheres via the spin coating method and were annealed at 600 °C. The processing step involved patterning and coating a $TiO_2$ sol–gel. The photocatalytic performance was analyzed using an UV–visible spectrophotometer. Within 20 min, a high catalytic efficiency (98% removal) with a 20-time faster decomposition rate of the malachite green (MG) solution than that of the nonpatterned $TiO_2$ was obtained from the patterned $TiO_2$ with 400 nm sized PS due to the large surface area. In addition, the phenol in the water removed as much as 50% within 2 h with the same photocatalyst, which was expected to be one of the strong candidates to be applied to the next generation of photocatalysts for water purification.

**Keywords:** titanium dioxide; photocatalyst; polystyrene sphere; patterned structure; dye and phenol removal

## 1. Introduction

The degradation of toxic materials from the biosphere is a multibillion dollar industry [1]. The most common methods rely on the use of high temperatures to accomplish the degradation, but such processes are expensive and must be carefully controlled, and the removal of effluent gases is challenging [2–5]. Textile, paper, dye, and pharmaceutical manufacturing facilities can contaminate water with residual dyes, which are mostly organic pollutants that are highly toxic and hazardous to humans and animals, and thus the removal of these organic contaminants prior to discharge into the environment is essential. Various insitu treatment methods based on bioremediation and electroreduction can be used, but each has inherent limitations [6]. Recently, a number of advanced, photocatalytic oxidation processes have been investigated, some of which appear to be promising alternatives, especially those that use natural sunlight as an energy input source [7–9]. In 1972, Fujishima and Honda demonstrated the potential of titanium dioxide ($TiO_2$) semiconductor materials to split water into hydrogen and oxygen in a photoelectrochemical cell [10]. Their work triggered the development of semiconductor photocatalysis for a wide range of environmental and energy applications [11]. $TiO_2$ is an example of a solid state semiconductor, characterized by two "bands" of closely spaced electronic energy levels known as the valence and conduction bands, which are respectively analogous to the lowest unoccupied molecular orbital (LUMO) and the highest occupied molecular orbital (HOMO). When electrons are promoted from the valence band to the conduction band, they become delocalized, and the substrate can conduct electricity (Scheme 1).

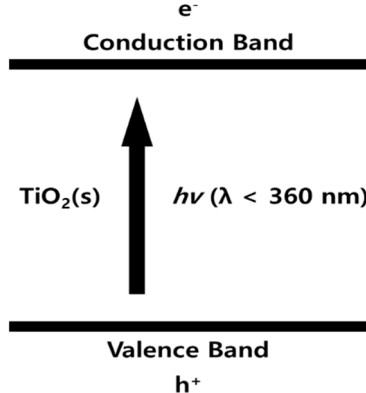

**Scheme 1.** Band energy structure of a typical solid state semiconductor, $TiO_2$.

Semiconductors have band gaps of intermediate energies between the conductor and insulator, and they can conduct if that band gap energy is supplied, thus elevating electrons from the valence into the conduction band. For anatase (a specific phase of $TiO_2$) and colloidal size regimes, this gap is roughly 320 kJmol$^{-1}$. Consequently, electronic excitation can be achieved by photons of light with $\lambda$ being $\leq$ 360 nm [12]. Therefore, $TiO_2$ is a particularly important photocatalyst due to its oxidizing power, nontoxicity, and long-term photostability. Not only the electronic properties of a material but also its structure and morphology can have aconsiderable influence on its photocatalytic performance. Most photocatalytic tests were carried out both with $TiO_2$ nanoparticle suspensions (or 2D $TiO_2$ thin films), which are done under open circuit conditions, indicating that electron and hole transfer occur from the same particle, and in a photoelectrochemical two-electrode configuration where $TiO_2$ is generally used as a photoanode together with an inert or catalytic cathode such as Pt, C, etc. Over the past years, this is why nanotube geometries (particularly anodic $TiO_2$ nanotube layers) have gained a lot of interest due to various potential advantages. In recent years particularly, 1D structures such as nanowires and nanotubes have also received great attention for their use as a photoelectrode [13–16].

The most important factors that influence the photocatalysis of $TiO_2$ nanotubes are the crystallinity, length, and diameter of the tubes, together with compositional effects. In early reports, it was demonstrated that the nanotube layers can have a higher efficiency than comparable compacted nanoparticle layers [17]. As for particles and as expected from a point of zero charge of $TiO_2$ of approximately 6–7, for acidic pH typically a better adsorption of, for example, COO$^-$ containing molecules (for example dyes) was observed, and typically at least slightly increased photocatalytic kinetics was observed [18]. As a powder particle, however, its use is somewhat inconvenient as it must be separated from the water in a slurry system after the photocatalytic reaction is complete. Accordingly, the development of a different method to apply $TiO_2$ coatings to various substrates is being pursued by researchers around the world [19–21].

Since the basic photocatalytic method investigated in our laboratory that used a colloidal suspension of titanium dioxide ($TiO_2$) has already been published [22], the main purpose of this study involves the synthesis of nanosized polystyrene (PS) patterned $TiO_2$ thin films to enhance both the surface area and the photocatalytic performance.

## 2. Results and Discussion

The synthesized PS colloidal monolayer patterned titanium dioxide films (with diameters of 400, 700, 1000, and 1300 nm, respectively) were characterized by field emission scanning electron microscopy (FE-SEM, Model JSM-7100 F) to evaluate their structures (Figure 1). The cross-sectional images (Figure 1e–h) show the monolayer patterned $TiO_2$ films on the PS spheres mask. The $TiO_2$ films with 1000 and 1300 nm sized patterns had well-ordered honeycomb-like structures (Figure 1c,d), but the $TiO_2$ films with 400 and 700 nm sized patterns had disordered structures (Figure 1a,b) and high surface charges. Therefore, smaller PS spheres had more difficulties aligning in the monolayer than the

larger PS spheres. However, as these samples had relatively higher surface are as compared with those of the bigger PS spheres, higher catalytic activity was expected.

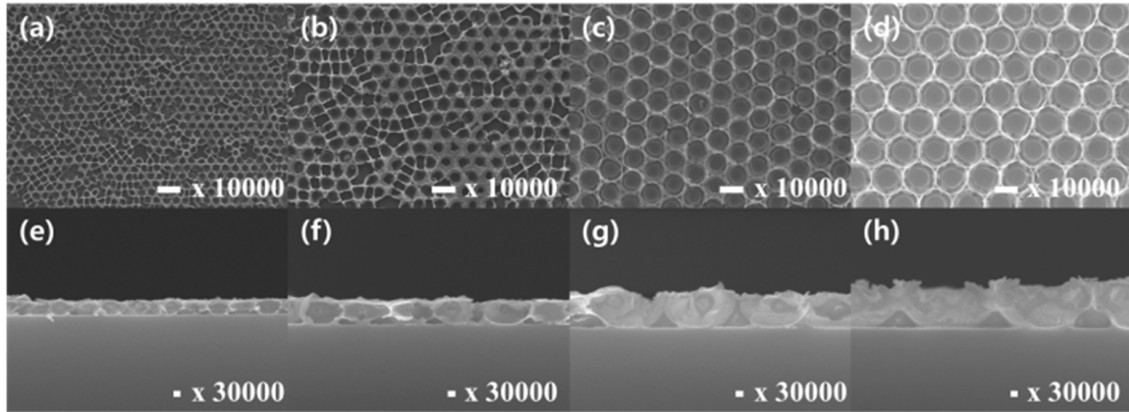

**Figure 1.** FE-SEM images of the top and cross-sectional views of the fabricated structures from the polystyrene(PS) patterned $TiO_2$ films with 400 nm (**a**,**e**), 700 nm (**b**,**f**), 1000 nm (**c**,**g**) and 1300 nm (**d**,**h**). Scale bars were 1000 nm for (**a**–**d**), and 3000 nm for (**e**–**h**).

The interesting thing is that the morphology of the PS-patterned $TiO_2$ films, based on the top-view SEM, is similar to that of anodic $TiO_2$ nanotubes [15]. Moreover, the AFM images (Figure 2) of the PS-patterned $TiO_2$ films are similar to that of $TiO_2$ nanotubes [16]. In this paper, therefore, the pros and cons of the PS-patterned $TiO_2$ films are described and then compared to those of anodic $TiO_2$ nanotubes. To mention briefly, the main advantage of this work is that it can control the height and distance of the pore size in PS-patterned $TiO_2$ films by varying the diameter of the PS sphere and adopting the oxygen plasma etching technique. However, compared to the anodic $TiO_2$ nanotubes, the PS-patterned $TiO_2$ films have limitations in the production of very thick $TiO_2$ films, which are inferior in terms of the specific surface area.

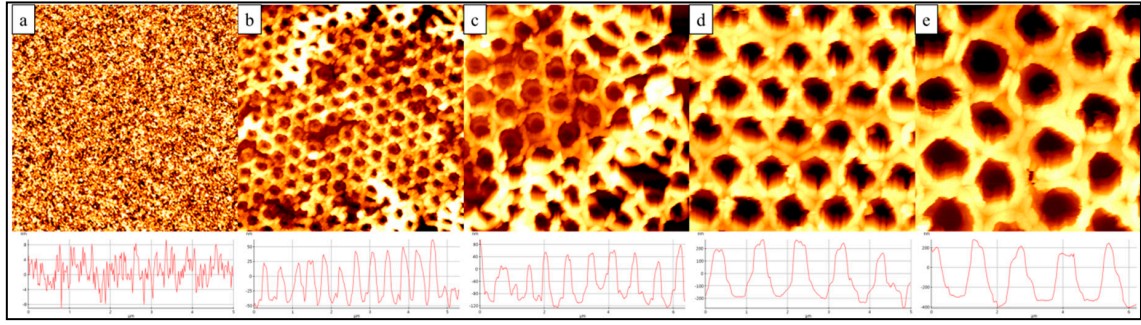

**Figure 2.** Atomicforce microscopy(AFM) images with line profiles of nonpatterned $TiO_2$ films (**a**) and PS-patterned $TiO_2$ films with diameters of 400 nm (**b**), 700 nm (**c**), 1000 nm (**d**) and 1300 nm (**e**).

Figure 2 depicts atomicforce microscopy (AFM) images, taken using an atomic force microscope (Park NX10-Atomic Force Microscope) of both nonpatterned PS-patterned $TiO_2$ thin films and the patterned $TiO_2$ using the PS spheres mask with diameters of 400, 700, 1000, and 1300 nm with the same silicon wafer. A surface morphology similar to that of FE-SEM was obtained. However, AFM images with line profiles show a different average depth of both the nonpatterned and PS-patterned $TiO_2$ thin films, as shown in the line profile range of 50, 100, 200, 400, and 500 nm, respectively (Figure 2). With these line profiles, the surface areas of both nonpatterned and PS-patterned $TiO_2$ thin films were obtained using an AFM-supported computer program. The calculated values for both PS-patterned (400, 700, 1000, and 1300 nm) and nonpatterned $TiO_2$ thin films were 67.8, 50.8, 43.4, 35.2, and 26.5 $nm^2$, respectively. Since it was very difficult to measure the Brunauer–Emmett–Teller (BET) surface area

of the 2D-thin films (especially for the nonpatterned and small-sized PS-patterned TiO$_2$ thin films) with very small pore volumes rather than those of 3D-powders, in this study, the surface area was measured using AFM instead of the BET method.

Figure 3 shows the crystallinity of the patterned TiO$_2$ film with a 400 nm size PS taken using X-ray diffractometer (XRD, D/Max Ultima III, Rigaku Corporation). The XRD patterns were found to have a main peak at 2θ = 25.2°, presumably due to the anatase (101) plane (JCPDS 21-1272), suggesting that all films mostly have a {101} facet, and there will be no influence of catalytic activity change on catalyst type. Peaks corresponding to the rutile and brookite phases were also not detected [23], and thus a high catalytic activity is expected.

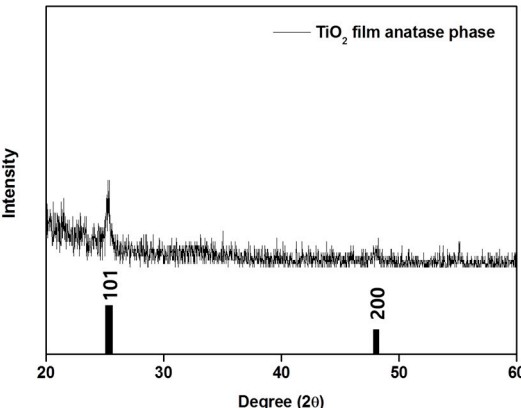

**Figure 3.** XRD pattern of the 400 nm size PS-patterned TiO$_2$ film.

X-ray photoelectron spectroscopy (XPS) measurements were made in order to observe the chemical state changes of PS-patterned TiO$_2$ thin films during the manufacturing process. In Figure 4, Ti 2p$_{1/2}$ was observed at ~463 eV, and Ti 2p$_{3/2}$ was observed at ~458 eV. For the PS-patterned TiO$_2$ thin films (400, 700, 1000, and 1300 nm), a chemical shift also occurred. However, it appears there was no distinct order and no binding energy difference for all samples [24]. Accordingly, there was no difference due to the defects of TiO$_2$ inducing by oxygen vacancy when making the films. Therefore, we concluded that all PS-patterned TiO$_2$ thin films were observed only for the Ti$^{4+}$ chemical state. In particular, only small changes in relative intensity (400 > 700 > 1000 > 1300 nm) of the TiO$_2$ films with PS patterns were observed. This means that the intensity of XPS spectra changes very little; therefore, it is suggested that as the PS size decreases, the amount of the TiO$_2$ increases and the intensity of the XPS spectrum increases when decreasing the PS sizes. In other word, the same sequence such as the relative intensity (400 > 700 > 1000 > 1300 nm) of PS-patterned TiO$_2$ thin films was obtained because of the same mean free path of the electron in the TiO$_2$ film layers under the same energy ranges of X-ray photoelectrons, even though the samples have different thickness. These results would be closely related with the values of the surface area.

For testing of photocatalytic activity, amalachite green (MG: C$_{23}$H$_{25}$ClN$_2$) solution was prepared by dissolving 0.001 g of MG in 1000 mL of distilled water. The concentration of this solution was $2.74 \times 10^{-6}$ M. The prepared PS-patterned TiO$_2$ films on $20 \times 20$ mm$^2$ sized silicon wafers were placed in a two-inch plastic dish, which was filled with 10 mL of MG solution. The plastic dish was placed in a black box and irradiated using ultraviolet (UV) light with a wavelength of 254 nm. Using an UV–visible spectrometer, the absorption spectrum of the MG solution was measured for 120 min, and the results are plotted in Figure 5. The inset of Figure 5 depicts the initial decomposition behavior (enlarged plot of the Figure 5 from 0 to 25 min) of the MG solution with 400–1300 nm PS-patterned TiO$_2$ films. As shown in the inset of Figure 5, within 20 min, all the PS-patterned photocatalysts tested in this study decomposed at least 95% or more of the MG molecules present. Complete decomposition of the MG solution occurred after 120 min. However, use of the nonpatterned TiO$_2$ films resulted in

only 63% degradation of the MG solution after 120 min. Of the catalysts tested, the highest efficiency (98% removal) resulted from use of the patterned TiO$_2$ with 400 nm size PS, perhaps due to the large surface area. This result clearly shows that PS patterning can increase the photocatalytic efficiency as much as 20 times higher than that of nonpatterned TiO$_2$ film in the initial 20 min. The introduction of porous channels into macroporous TiO$_2$ increased the photocatalytic activity due to the minimization of intradiffusion resistance and the enhancement of photoabsorption efficiency [25]. In the 3D materials, the macrochannels can serve as effective paths for light and reactant transportation. This allows the UV light to penetrate more deeply inside the porous TiO$_2$ films, resulting in the enhancement of degradation efficiency and rate of MG by TiO$_2$ photocatalysts fabricated with PS patterning with a different diameter [26].

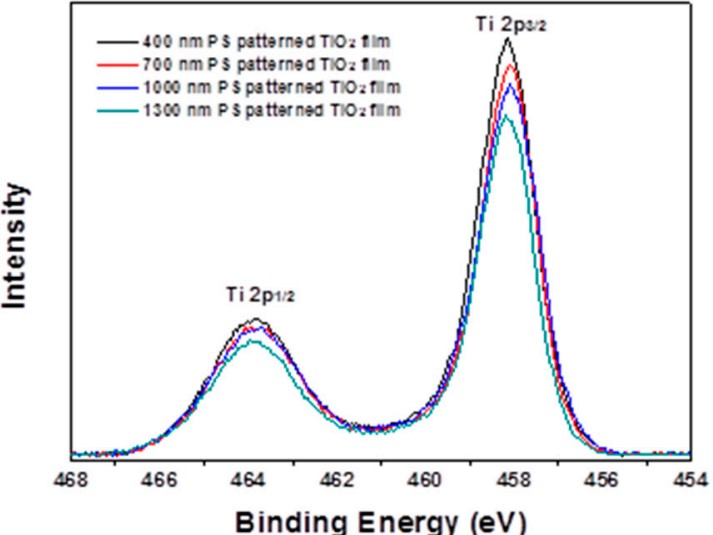

**Figure 4.** High resolution Ti 2p X-ray photoelectron (XP) spectra of the TiO$_2$ photocatalysts synthesized with 400 nm, 700 nm, 1000 nm, and 1300 nm patterned TiO$_2$ films.

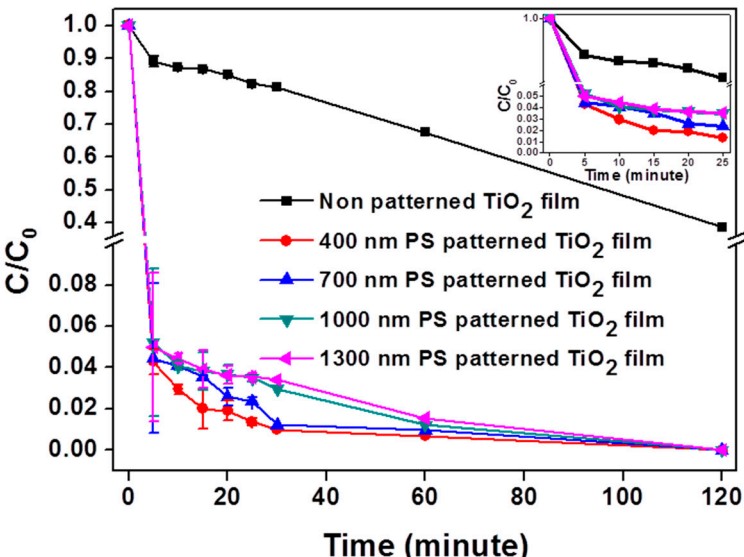

**Figure 5.** Photocatalytic degradation of malachite green (MG) with the nonpatterned TiO$_2$ film and PS-patterned TiO$_2$ films. Inset shows photocatalytic degradation of MG for initial 25 min.

As we mentioned before, there are many factors that influence the photocatalysis of both $TiO_2$ nanotubes and particles (in this case the PS-patterned $TiO_2$ films). Among them, we will consider the length, diameter, together with surface area. For example, a report of either a maximum in the photocatalytic activity for tube layer thicknesses around 3–7 μm [27,28] or the absence of an influence of the tube length [29] was announced. In addition, there is a discrepancy for the influence of tube diameter. For example, one report announced no significant influence [30,31], but the other report showed maximum photocatalytic efficiency at around −100 nm [28,32], or other trends [33]. These discrepancies can be attributed to the fact that it is very difficult to vary tube length independently from tube diameter. Besides $TiO_2$ nanotubes, there are also reports about other forms of self-organized structures such as self-organized mesoporous $TiO_2$ [34]. These structures, named as "titaniamesosponge"(TMS) or "nanochannelar"structures, can contain significant crystallinity (anatase and anatase/rutile) and when annealed can show an enhanced photocatalytic activity as compared to P25 layers, depending on layer thickness and annealing conditions (i.e., different surface area) [35]. Since our PS-patterned $TiO_2$ films have a similarity to $TiO_2$ nanotubes, both parameters (such as diameter, thickness layers, morphology/crystallinity, and annealing at 600 °C) will affect the photocatalytic activities, resulting in a 20-time faster decomposition rate of MG dye solution with 400 nm PS-patterned $TiO_2$ film compared with that of nonpatterned $TiO_2$. However, a more detailed kinetic study including detection of key radicals such as O(1D) and OH· radicals is highly desirable to clarify our amazing result. It is well known that the average lifetime of OH radical (τOH) in an ambient atmospheric condition is around 0.01–1 s [36], which is affected by the concentration of reactive gas components such as ozone, VOCs, and NOx.

From an application viewpoint, the most important reactions are the transfer of valence band electronsto $H_2O$, $H^+$, or $O_2$ and the transfer of holes to $H_2O$, $OH^-$, or organic species. If we consider an aqueous environment, then the transfer of conduction band electrons may lead to the production of $H_2$. For the valence band holes, except for a reaction with $OH^-$ or $H_2O$ to form $O_2$, OH· radical formation may also occur and is often the desired reaction for pollution degradation. In this case, formed OH· radicals are able to virtually decompose all organics to $CO_2$ + $H_2O$. Nevertheless, if the $H_2O$ concentration is comparably small, valence band holes may also be transferred directly to the organics and lead to their decomposition. Theoretically, therefore, maximum efficiency for the photocatalytic reaction is when all charge carriers react with the species from the surroundings rather than recombine.

To understand this in detail, the efficiency of MG dye decomposition and the surface area of the PS-patterned $TiO_2$ thin films were compared after irradiating 254 nm UV light into the PS-patterned thin films for 120 min (Figure 6). The surface area was calculated by AFM, as shown in the part of Figure 2, to allow for an accurate comparison of the patterns according to the PS size of 400, 700, 1000, and 1300 nm. Based on Figure 5, the maximum degradation efficiencies, such as 97.8%, 96.6%, 96.0%, 95.6%, and 63.0% (not shown in Figure 6), were obtained from both PS-patterned (400, 700, 1000, and 1300 nm) and nonpatterned $TiO_2$ thin films, respectively. As a result of the surface area calculation, from Figure 6, we realized that the degradation efficiency improved in proportion to the surface area. This means that there will be a close relationship between the photocatalytic efficiency and the surface area, and a detailed mechanism supporting this relationship should highly be desirable for clarifying results [37,38].

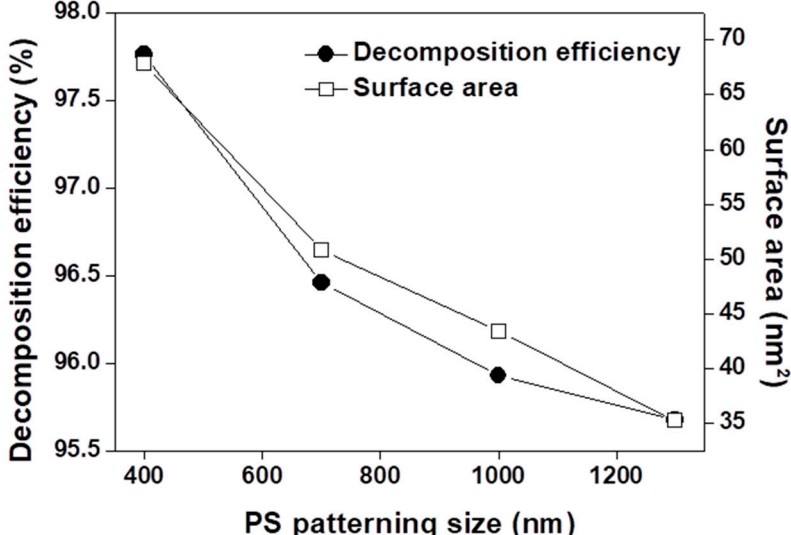

**Figure 6.** The initial photodegradation efficiency of PS-patterned films irradiated with a UV lamp for 25 min and the variations of surface area among the PS-patterned films.

Based on our experimental data, especially shown in the inset of Figure 5, the reaction constants of both nonpatterned $TiO_2$ and PS-patterned $TiO_2$ were calculated using the following equation by assuming the first-order kinetics.

$$\text{The reaction rate} = -\frac{d[C]}{dt} = k[C]$$
$$\frac{1}{k}\int_{[C]_0}^{[C]_t}\frac{1}{[C]}\times d[C] = -\int_0^t dt \tag{1}$$
$$\frac{1}{k}\times(\ln[C]-\ln[C]_0) = -t$$

$$\ln(C/C_0) = -kt \tag{2}$$

$$C = C_0\exp(-kt) \tag{3}$$

where $C_0$ and $C$ are the initial concentration and the concentration of MG solution, respectively; $t$ is the photoreaction time; and $k$ is the reaction constant. Equation (3) has the form of an exponential decay (similar to theinset of Figure 5). A common feature of all first-order reactions, therefore, is that the concentration of the reactant (i.e., the MG photodegradation in this study) decays exponentially with time. However, there are different exponential decay curves depending on $k$ values. The greater the rate constant, the more rapid is the decay curve. Based on the Equation (2), we can get the rate constants ($k_1$/min) from each decay curve. Figure 7a shows a plot of $\ln(C/C_0)$ versus the photocatalytic reaction time ($t$) for both the nonpatterned $TiO_2$ film and the 400 nm PS-patterned $TiO_2$ film irradiated with 254 nm UV light for 25 min under the same experimental condition. From the slopes of each plot, the rate constants ($k_{obs}$/min) of 0.0064 and 0.140 were obtained. From the similar experiments on the PS-patterned (700, 1000, and 1300 nm) $TiO_2$ films, the rate constants ($k_{obs}$/min) with 0.117, 0.102, and 0.101 were also obtained, respectively (not shown in Figure 7a). Figure 7b shows variation of the rate constant with the PS diameter. In increasing the PS diameter from 400 to 1300 nm, there was a decreasing tendency of rate constant from 0.14 to 0.10. The perfect linearity indicates the first-order kinetics with the first-order rate constant ($k_1$/min), but the experimentally obtained data in this work have large deviations, suggesting a pseudo first-order kinetics (not perfect first-order reaction) with pseudo first-order rate constants ($k_{obs}$/min). Therefore, Figure 7 provides us small hints regarding the effects of the initial MG concentration on the removal efficiency. First, the $TiO_2$ thin film patterned with 400 nm PS beads was observed to decompose the MG dye the most rapidly (20 times faster than nonpatterned $TiO_2$ thin film). Second, the photocatalytic degradation of MG in both the 400 nm

size PS-patterned and nonpatterned TiO$_2$ thin films exhibits pseudo first-order kinetics with different rate constants (k$_{obs}$) [39]. This means that the kinetics of photocatalytic degradation of MG in both nonpatterned and PS-patterned TiO$_2$ thin films were the same, but there were large differences in value (maximum 22 times) of the rate constants between nonpatterned and PS-patterned TiO$_2$ thin films. It is very important to note that a rate law is established experimentally and cannot in general be inferred from the chemical equation for the reaction. Moreover, the reaction order can only be determined experimentally. Since we did not measure either the half-life (which is independent of reactant concentration for the first-order kinetics) or the time constant (i.e., lifetime; the longer the time constant of the first-order reaction, the slower the decay and the longer the reaction survives) by a laser spectroscopic technique, further experimentation is highly desirable for determining the exact reaction kinetics.

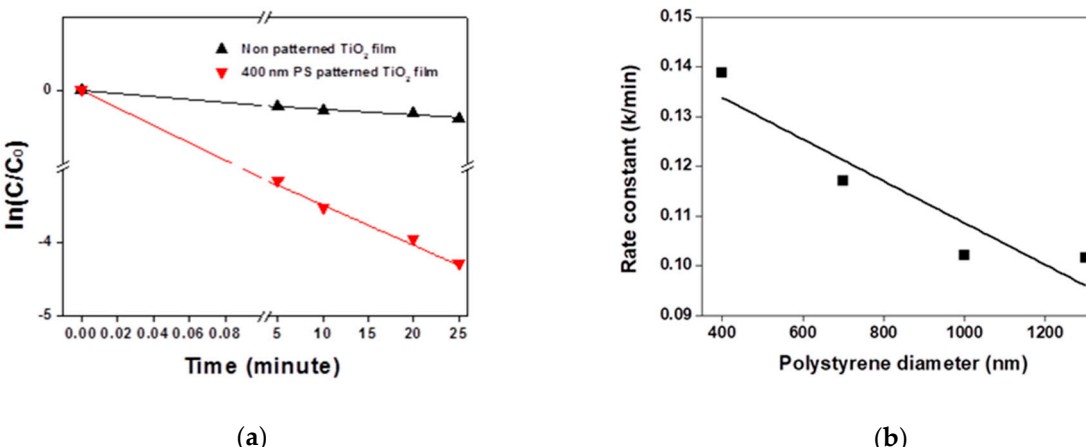

(**a**)            (**b**)

**Figure 7.** (**a**) Effect of initial MG concentration on the removal efficiency: the plot of ln(C/C$_0$) versus irradiation time for the nonpatterned TiO$_2$ film and the 400 nm PS-patterned TiO$_2$ film for 25 min. (**b**) Variation of rate constant with PS diameter. The perfect linearity indicates the first-order kinetics with the first-order rate constant (k$_1$/min), but the experimental data have deviations, suggesting a pseudofirst-order rate constants (k$_{obs}$/min).

For environmental application such as water purification [37], a phenol degradation experiment was also performed using the best catalyst to determine the ability of the patterned TiO$_2$ films to purify water. Figure 8 depicts the photodegradation of phenol in the presence of 400 nm PS-patterned TiO$_2$ films. The photocatalytic performance measurements were carried out by exposing the 400 nm PS-patterned TiO$_2$ film to UV light with a wavelength of 254 nm while soaking it in a 200 ppm solution of phenol solution for 120 min. Phenol exposed to UV light for 120 min was decomposed until a concentration of 100 ppm was reached (50% removal within 2 h), suggesting that the proposed PS-patterned TiO$_2$ thin film will be one of the possible strong candidates for new types of photocatalysts for both dye decomposition and water purification. This suggests that it is especially important for future applications in environmental purification processes to develop PS-patterned TiO$_2$ films on non flat supports such as 3D meshes/grids, spheres, or others with good adhesion [40].

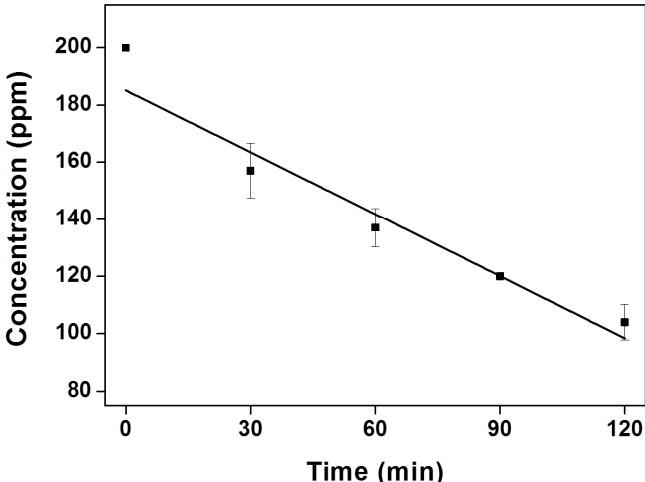

**Figure 8.** Photocatalytic degradation of phenol in the presence of the 400 nm PS-patterned $TiO_2$ film.

## 3. Materials and Methods

### 3.1. Preparation of Polystyrene (PS) Spheres

The size of the polystyrene (PS) spheres affects the reaction time during the dispersion and polymerization of styrene. With increasing reaction time, the size of the PS spheres increased proportionally (400, 700, 1000, and 1300 nm). An ordered PS sphere monolayer was generated using the conventional air–water interface mediated method. This indicated that blowing led to an improvement in the crystal domain size by facilitating recrystallization during the self-assembly process. Before growing, the apertures of the PS monolayer were modified by thermal treatment in a tube furnace, which ensures uniformity throughout the sample and allows for good control over the aperture size by an adjustment of the adhesion time. Adhesion was achieved by using a hot plate at 120 °C. Field emission scanning electron microscopy (FE-SEM, JEOL Corp., Model JSM-7100 F, Tokyo, Japan) images (Figure 1) of the synthesized PS spheres show highly uniform diameters.

### 3.2. Preparation of the $TiO_2$ Sol–Gel

The $TiO_2$ sol–gel catalysts (pH= 2.2 and 5.7 wt%) were prepared by mixing titanium hydroxide oxide ($Ti(OH)_2O$) with hydrochloric acid (HCl) at a molar ratio of 1:3 in ethanol. The solution was stirred at 80 °C for 2 h, and concentrated in vacuum to give a white slurry. The slurry was then kept for 1 day at 80 °C in an electric oven, after which it turned into a yellow powder, which was annealed in the furnace at 600 °C for 17 min.

### 3.3. Synthesis of Nonpatterned and PS-Patterned $TiO_2$ Thin Films

Silicon (100) wafers (20 x 20 $mm^2$) were cleaned by sonication with acetone (10 min), ethyl alcohol (10 min), and distilled water (10 min), followed by drying with $N_2$ gas. The cleaned silicon wafers were treated with oxygen plasma to enhance the adhesion of variously sized PS spheres at 100 Watts, 100 sccm, and 5 min. PS beads of various sizes were arranged on the silicon wafers to increase the surface area, using 400, 700, 1000, and 1300 nm for a monolayer coating as well as the PS spheres. After this, 200 μL of $TiO_2$ sol–gel was dropped onto the PS-patterned silicon wafers and allowed to sit for 1 min. Afterward, the PS substrates were coated using the spin coating method (3000 rpm, 30 s) and annealed in a furnace at 600 °C for 6 h to provide PS-patterned $TiO_2$ films. Nonpatterned $TiO_2$ films were produced in the same way as PS-patterned $TiO_2$ films but without a PS monolayer coating. The detailed deposition process of the nonpatterned $TiO_2$ films was already published elsewhere [16].

*3.4. The Methods of Photocatalytic Activity Measurements*

Catalytic degradations of both a malachite green (MG) solution and phenol in water by various PS-patterned $TiO_2$ films were undertaken to evaluate the catalytic efficiency of the films, which was monitored using UV–visible absorption spectroscopy (UV-3600 Plus UV-VIS-NIR Spectrophotometer: SHIMADZU, Kyoto, Japan). For testing of photocatalytic activity, a $MG(C_{23}H_{25}ClN_2)$ solution with a concentration of $2.74 \times 10^{-6}$ M was prepared. The prepared PS-patterned $TiO_2$ films on $20 \times 20$ mm$^2$ sized silicon wafers were placed in a two-inch plastic dish, which was filled with 10 mL of MG solution. The plastic dish was placed in a black box and irradiated using ultraviolet (UV) light with a wavelength of 254 nm and a power of 10 W (Spectroline, NY, USA). Using an UV–visible spectrometer, the absorption spectrum of the MG solution was measured in the wavelength range between 400 and 800 nm for 120 min. The phenol degradation measurements were also carried out by exposing the 400 nm PS-patterned $TiO_2$ film to UV light with a wavelength of 254 nm while soaking it in a 200 ppm solution of phenol solution for 120 min (Sigma-Aldrich, St. Louis, USA).

## 4. Conclusions

In this study the polystyrene (PS)-patterned $TiO_2$ thin films were synthesized to enhance both the surface area and the photocatalytic performance. In the photocatalytic efficiency test, all patterned $TiO_2$ thin films decomposed MG dye almost 100% while the nonpatterned $TiO_2$ thin films decomposed MG dye by about 63% in 120 min, and a 20-time higher photocatalytic decomposition rate of MG dye was obtained within 20 min with 400 nm PS-patterned $TiO_2$ thin film compared with that of nonpatterned $TiO_2$ film. For water purification, a phenol degradation experiment was also performed using the same photocatalyst, resulting in the removal of phenol in the water as much as 50% within 2 h.

However, the kinetics of MG dye decomposition shows the same mechanism (pseudo first-order kinetics). All the $TiO_2$ films tested were capable of decomposing harmful material such as phenol into the patterned film, but the highest performing thin film was for the 400 nm PS sphere patterned thin film. Since the process used to layer the $TiO_2$ onto the PS spheres was cheaper and easier than previous methods, and it was also amenable to large area substrates, future studies to further develop this technology are highly expected. Because this is especially important for future applications in environmental purification processes, we have to develop a technique on preparing a catalyst onto both flat and non flat supports with good adhesion.

**Author Contributions:** Conceptualization, H.J.S and J.-H.B.; methodology, Y.H.N.; software, J.W.L; validation, H.J.S., J.W.L. and J.-H.B.; formal analysis, H.J.S.; investigation, Y.H.N.; resources, J.-H.B.; data curation, H.J.S.; writing—original draft preparation, H.J.S.; writing—review and editing, J.-H.B.; visualization, J.W.L.; supervision, J.-H.B.; project administration, J.-H.B.; funding acquisition, J.-H.B. All authors have read and agreed to the published version of the manuscript.

**Funding:** This work was supported by the National Research Foundation of Korea (NRF) grant funded by the Korea government (MSIT) (2020R1A2C1011764).

**Conflicts of Interest:** The authors declare no conflict of interest.

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
