# Peer review of "Enhancement of Photocatalytic Activities with Nanosized Polystyrene Spheres Patterned Titanium Dioxide Films for Water Purification"

_catalysts, doi:10.3390/catal10080886_

Round 1

Reviewer 1 Report

This paper reporst on nano-sized polystyrene spheres patterned TiO2 films for photocatalytic water purification. The content of the present work is interesting and the study itself sounds good. Nevertheless, points below needs to be addressed before my approval for publication in Catalysts.

  1. Morphology of the PS-patterned TiO2 films, based on the top-view SEM, is similar to that of anodic TiO2 nanotubes (e.g., Lee et al Chemical Reviews 114 (2014), Macak et al Angewangte Chemie 44 (2005). The reviewer thinks that this fact is of high interest for the future potential readers and it should be mentioned in the paper. The authors should discuss pros and cons of their PS-patterned TiO2 films compared to that of anodic TiO2 nanotubes. Moreover, also AFM images of PS-patterned TiO2 films is similar to that of TiO2 nanotubes (e.g., Sopha et al, Materials Research Bulletin 103 (2018) 197-204).

  1. Why is silicon wafer used as a substrate? Is this advantageous in some way? Is it possible to prepare such TiO2 films on other substrates? Again, similar to anodic TiO2 nanotubes that can be prepared on whole range of substrates such as Si wafer (e.g., Motola et al Chemical Papers 73 (2019) 1163-1172), the authors should discuss pros and cons of their PS-patterned TiO2 films compared to anodic TiO2 nanotubes.

  1. Is it possible to prepare such PS-patterned TiO2 films on other non-flat supports? For instance 3D meshes/grids, spheres or other? This is especially important for future applications in environmental purification processes and authors should discuss this.
  2. Is it possible to prepare much thicker TiO2 films, i.e., in the range of hundreds of nm or even μm?

  1. A more detailed description of the photocatalytic activity measurements need to be added into the experimental part, e.g., what is the concentration of MG and phenol solution? Description of the light source used during measurements (i.e., intensity/power, wavelength range, etc.)

Author Response

Dear Referee,

Thank you very much for refereeing our paper and for giving us your kind corrections and suggestions as well as valuable comments. We have been accepted your fruitful comments, suggestions, and recommendations with all our heart and made our best to revise the manuscript (marked with red color for the corrected parts) with your advice. Thus, hereby, we prepared the answers for you as followings.

Referee 1

  1. Morphology of the PS-patterned TiO2 films, based on the top-view SEM, is similar to that of anodic TiO2 nanotubes (e.g., Lee et al Chemical Reviews 114 (2014), Macak et al Angewangte Chemie 44 (2005). The reviewer thinks that this fact is of high interest for the future potential readers and it should be mentioned in the paper. The authors should discuss pros and cons of their PS-patterned TiO2 films compared to that of anodic TiO2 nanotubes. Moreover, also AFM images of PS-patterned TiO2 films is similar to that of TiO2 nanotubes (e.g., Sopha et al, Materials Research Bulletin 103 (2018) 197-204).

 ⇒ Thank you very much for your valuable suggestions and comments. We added your suggestion into the revised paper, and described the pros and cons of their PS-patterned TiO2 films then compared to that of anodic TiO2 nanotubes. The detailed descriptions are given in the revised paper (especially on pages 2,3,5). Thus, pls. check it again.

  1. Why is silicon wafer used as a substrate? Is this advantageous in some way? Is it possible to prepare such TiO2 films on other substrates? Again, similar to anodic TiO2 nanotubes that can be prepared on whole range of substrates such as Si wafer (e.g., Motola et al Chemical Papers 73 (2019) 1163-1172), the authors should discuss pros and cons of their PS-patterned TiO2 films compared to anodic TiO2 nanotubes.

 ⇒ Regarding on the first question, there is no specific reason for using Si wafer in this paper. However, when using Si wafer as a substrate, there is an advantage of easy cutting of samples with proper size. Such periodic honeycomb patterned TiO2 films can be formed on various substrates such as common soda lime glass, indium tin oxide, sapphire, and other metallic substrates. However, polymer substrates are excluded because deformation occurs during high-temperature calcinations. In this experiment, PS-patterned TiO2 films were prepared by using colloidal lithography method through the modified Langmuir-Blodgett method. Such modified Langmuir-Blodgett method can produce large-area periodic nanostructure arrays with an ultrahigh throughput in cost-competitive ways.

[Please see the Nano Lett. 2015, 15, 4591−4598]

In addition, it can control the height and distance of pore size in PS-patterned TiO2 films by varying the diameter of PS sphere and adopting the oxygen plasma etching technique. However, compared to the anodic TiO2 nanotubes, the PS-patterned TiO2 films prepared in this study have limitations in the production of very thick TiO2 films, which are inferior in terms of specific surface area.

About the second and third questions: As we wrote above, yes it is possible to prepare TiO2 films on other substrates. We did prepare TiO2 films on max. 4 inch sized Si wafers. I think that it is basically possible to prepare TiO2 films on whole range of Si wafer. For example, a Chinese group, showing in above picture, reported 30 inch sized Si wafer using colloidal lithography technique.

  1. Is it possible to prepare such PS-patterned TiO2 films on other non-flat supports? For instance 3D meshes/grids, spheres or other? This is especially important for future applications in environmental purification processes and authors should discuss this.

⇒ Again, thank you very much for your kind suggestion. Yes, it is basically possible to prepare PS-patterned TiO2 films on other non-flat supports such as 3D meshes/grids, spheres or other. In this moment, however, we only tried to prepare PS-patterned TiO2 films on 3D meshes/grids. That’ why we have to develop a technique about preparing a catalyst onto non-flat supports such as 3D meshes/grids, spheres or other with good adhesion.

Since we fully agreed with your opinion, we added the following sentences in the revised paper. So, pls. check it up.

Because this is especially important for future applications in environmental purification processes, we have to develop a technique about preparing a catalyst onto both flat and non-flat supports with good adhesion.

  1. Is it possible to prepare much thicker TiO2 films, i.e., in the range of hundreds of nm or even μm?

     ⇒ Yes, it is possible to prepare much thicker TiO2 films up to 1 μm. But, it is strongly dependent on the diameter of PS beads as well as growth method of TiO2 films.

  1. A more detailed description of the photocatalytic activity measurements need to be added into the experimental part, e.g., what is the concentration of MG and phenol solution? Description of the light source used during measurements (i.e., intensity/power, wavelength range, etc.)

 ⇒ We upgraded the part of “3.4. The methods of photocatalytic activity measurements” with adding a more detailed description such as the concentrations of MG and phenol solutions as well as description of the light source used during measurements (i.e., intensity/power, wavelength range, etc.). Thus, pls. check it again.

Reviewer 2 Report

The manuscript describes a method to increase the surface area of TiO2 by adding them to in a pattern to a polysulfone surface. Polysulfone films of different sizes have been tested (from 400 to 1300 nm) with a non-patterned PS-surface as reference.

That TiO2 can be used to generate radicals after exposure of UV-light is hardly new: a quick search in the chemical database used at my university library indicates more than 1000 papers published since 2019 in this area.

The basic message in this manuscript is the fact that the larger area of TiO2 available in the reaction, the faster the degradation of organic compounds will be, which is a trivial observation. I think the work described in the manuscript is interesting, but would like to suggest an analysis of how many radicals can be generated by the different patterned films. The authors are asked to include this analysis in the revision.

1) Can the amount of generated radicals be measured or calculated based on the available catalytic surface for the different PS-films?
2) Can the concentration of radicals be related to the kinetics of malachite green degradation?

Author Response

Dear Referee,

Thank you very much for refereeing our paper and for giving us your kind corrections and suggestions as well as valuable comments. We have been accepted your fruitful comments, suggestions, and recommendations with all our heart and made our best to revise the manuscript (marked with red color for the corrected parts) with your advice. Thus, hereby, we prepared the answers for you as followings.

Referee 2

The manuscript describes a method to increase the surface area of TiO2 by adding them to in a pattern to a polysulfone surface. Polysulfone films of different sizes have been tested (from 400 to 1300 nm) with a non-patterned PS-surface as reference. That TiO2 can be used to generate radicals after exposure of UV-light is hardly new: a quick search in the chemical database used at my university library indicates more than 1000 papers published since 2019 in this area.

The basic message in this manuscript is the fact that the larger area of TiO2 available in the reaction, the faster the degradation of organic compounds will be, which is a trivial observation. I think the work described in the manuscript is interesting, but would like to suggest an analysis of how many radicals can be generated by the different patterned films. The authors are asked to include this analysis in the revision.

1) Can the amount of generated radicals be measured or calculated based on the available catalytic surface for the different PS-films?

⇒ Since we did not measure the ESR(electron spin resonance) spectra as well as decay time behavior of the generated radicals by a Laser spectroscopic technique, we can’t count the exact amounts of generated radicals as well as quantum yield. Therefore, we described the following sentences in the manuscript.

Since we did not measure either half life (that is independent of reactant concentration for the first-order kinetics) or time constant (lifetime – the longer the time constant of the first-order reaction, the slower the decay and the longer the reaction survive) by a Laser spectroscopic technique, further experiment is highly desirable for determining the exact reaction kinetics.

With your comments and suggestions, moreover, we added the following sentences as well.

More detailed kinetic study including detection of key radicals such as O(1D) and OH· radicals is highly desirable to clarify our amazing result. It is well known that the average lifetime of OH radical (τOH) in an ambient atmospheric condition is around 0.01∼1 s [36], which is affected by the concentration of reactive gas components such as ozone, VOCs, and NOx.

From an application viewpoint, the most important reactions are the transfer of valence band electrons to H2O, H+, or O2 and the transfer of holes to H2O, OH, or organic species. If we consider an aqueous environment, then the transfer of conduction band electrons may lead to the production of H2. For the valence band holes, except for a reaction with OH or H2O to form O2, also OH· radical formation may occur and is often the desired reaction, for pollution degradation. In this case, formed OH· radicals are able to virtually decompose all organics to CO2 + H2O. Nevertheless, if the H2O concentration is comparably small, valence band holes may also be transferred directly to the organics and lead to their decomposition. Theoretically, therefore, a maximum efficiency for the photocatalytic reaction is when all charge carriers react with the species from the surroundings rather than recombine.

In the future, anyhow, we will measure both characteristics to clarify more detailed kinetic mechanism.

2) Can the concentration of radicals be related to the kinetics of malachite green degradation?

 ⇒ Yes, the concentration of radicals is closely related to the kinetics of dye degradation. That’s why it is not exactly fit to the first order kinetics. For example, as for particles, and as expected from a point of zero charge of TiO2 of approximately 6−7, for acidic pH typically a better adsorption of, for example, COO- containing molecules (for example dyes) is observed, and typically at least slightly increased photocatalytic kinetics is observed.

3) It needs to improve the parts of both Introduction and Conclusions as well as the English spell check.

⇒ We upgraded the parts of both Introduction and Conclusions. Also, we checked the English spell through the manuscript. Thus, pls. check it again.

Round 2

Reviewer 2 Report

The paper has been improved.